



# sedInterFoam 1.0: a three-phase numerical model for sediment transport applications with free surfaces

Antoine Mathieu[1], Yeulwoo Kim[2], Tian-Jian Hsu[1], Cyrille Bonamy[3], and Julien Chauchat[3]

[1]Center for Applied Coastal Research, University of Delaware, Newark, DE 19716, USA
[2]Department of Sustainable Engineering, Pukyong National University, Busan 48513, Republic of Korea
[3]LEGI, University of Grenoble Alpes, G-INP, CNRS, 38000 Grenoble, France

**Correspondence:** Antoine Mathieu (amathieu@udel.edu)

**Abstract.** In this paper, an Eulerian two-phase-flow model sedFoam is extended to include an air phase together with the water and sediment phases. The numerical model called sedInterFoam is implemented using the open source library OpenFOAM. SedInterFoam includes the previous features of sedFoam for sediment transport modeling and also solves the air/water interface using the volume of fluid method coupled with the waves2Foam toolbox for free surface wave generation and absorption. Using sedInterFoam, four test cases are successfully reproduced to validate the free-surface evolution algorithm implementation, mass conservation of sediment and fluid phases, predictive capabilities and demonstrate its potential in modelling a broader range of coastal applications with sediment transport dominated by surface waves.

## 1 Introduction

Sediment transport is the main driver of morphological changes in the coastal and fluvial environments (Sherwood et al., 2022). Understanding and modelling the physical processes involved in the movement of sediment particles forced by a free surface flow is a major issue. Accounting for and integrating the complex mechanisms involved in sediment transport in numerical models such as interactions between solid particles and fluid turbulence, intergranular interactions, and surface wave-driven processes is a critical step to improve and enhance the existing model capabilities.

In the conventional single-phase modelling approach, Navier-Stokes equations are only solved for the fluid phase, sediment transport is related to the bottom shear stress with bed-load suspended-load formulations and the resulting morphological evolution is calculated using the Exner equation (Jacobsen and J., 2014; Baykal et al., 2017). This modeling methodology has several limitations, although it is widely used because of its simplicity and computational efficiency. One of the major limitations is related to the empirical relation between horizontal/vertical sediment fluxes and the bottom shear stress. Indeed, most of these relations are valid for steady or mild transient flows and relatively coarse sediments in controlled laboratory environment. As a result, a large discrepancy is observed between predicted and measured sediment flux for given wave time series (Yu et al., 2012; van der A et al., 2013) suggesting that sediment flux solely parameterized by the bottom shear stress is incomplete. Another major limitation of a conventional single-phase model is its inability to simulate onset of scour due to piping (Sumer et al., 2001), which involves gap formation and tunneling due to seepage flow driven by an upstream-downstream pressure gradient.





To overcome these limitations, the Eulerian two-phase modelling approach which encompasses most of the physical mechanisms involved in the coupling between water and particles has been actively developed in the last two decades (Dong and Zhang, 2002; Hsu et al., 2004; Amoudry, 2014; Lee et al., 2016; Chauchat and Guillou, 2008; Chauchat, 2018; Mathieu et al., 2021). In the Eulerian two-phase flow approach, both the carrier phase and the dispersed sediment phase composed of the particles are seen as interpenetrating continua. Coupling between the two phases is modeled using spatial-averaged interaction forces between the fluid and the particles (*e.g.* buoyancy, drag, added-mass) and particle-particle interactions can be included in the solid-phase stress closure models. As demonstrated in Tsai et al. (2022), an Eulerian two-phase model is able to simulate scour onset underneath a 2D pipeline due to seapage flow driven by a upstream-downstream pressure difference.

To make the scientific community benefit from the Eulerian two-phase model for sediment transport applications, an open source solver called sedFoam (Chauchat et al., 2017; Cheng et al., 2017) implemented using the CFD library OpenFOAM (Jasak and Uroić, 2020) has been developed. The solver includes the most recent developments for intergranular stress modeling such as the kinetic theory for granular flows (Chassagne et al., 2023) and the $\mu(I)$ rheology (Boyer et al., 2011), turbulence closure models such as mixing length (Revil-Baudard and Chauchat, 2013), $k - \varepsilon$ (Hsu et al., 2004), $k - \omega$ (Amoudry, 2014; Nagel et al., 2020), and large-eddy simulation (Mathieu et al., 2021) models. SedFoam has been validated using many benchmarks (Chauchat et al., 2017) and successfully applied to various practical configurations such as scour applications (Mathieu et al., 2019; Nagel et al., 2020; Tsai et al., 2022), ripple migration and geometry evolution (Salimi-Tarazouj et al., 2021a, b), sheet flow in an oscillatory boundary layer (Delisle et al., 2022; Mathieu et al., 2022) or immersed granular avalanches (Montellà et al., 2021, 2023).

Although two-phase flow models allowed to significantly improve our knowledge of sediment transport processes in the coastal and fluvial environments, this kind of modelling methodology is only applicable for configurations where free surface effects are negligible or under simplifying assumptions such as sediment transport in an oscillating water tunnel rather than under surface waves. Indeed, it is critical to include the capability to solve the propagation and breaking of surface waves in order to reproduce more realistic cross-shore sediment transport and morphodynamics including the swash zone. To do so, resolving a third separated gaseous phase and implementing a numerical algorithm to resolve the evolution of the free surface is necessary. Kim et al. (2018) were the first to tackle this problem by combining an early version of sedFoam, the free-surface solver interFoam (Klostermann et al., 2013) and a library to generate free-surface waves waves2Foam (Jacobsen et al., 2012). With their solver, they successfully predicted sheet flow under monochromatic non-breaking waves by comparing numerical results with experimental data from Dohmen-Janssen and Hanes (2002) and studied sheet flow near a sand bar under near-breaking surface waves (Kim et al., 2019, 2021). However, the numerical implementation of the model prevented the sediment to be present in the air phase and, as a consequence, the sediment bed had to be located far away from the free-surface. Despite the significant advances made possible by adding a third gaseous phase in their model, this limitation prevented to simulate configurations for which there are strong interactions between air, water and sediment such as in the swash zone. More recently, Lee et al. (2019) proposed a three-phase model similar to the one presented by Kim et al. (2018) using an equivalent but more general set of equation that allows sediment in both air and water phases.





To provide an open-source model for nearshore sediment transport applications, the purpose of this work is to present and
validate the open source solver sedInterFoam, an extension of the two-phase model sedFoam to simulate sediment transport
with a free surface and for intense interactions in the swash zone. Benefiting from the earlier work of Kim et al. (2018) in
combining sedFoam, interFoam and the wave generation library waves2Foam and the set of equations proposed by Lee et al.
(2019), sedInterFoam is able to simulate configurations for which sediment is present in both air and water. The model is first
presented in section 2. Then, model implementation and solution algorithm are detailed in section 3. In section 4, the model is
validated using benchmarks and applied to configurations related to cross-shore sediment transport and beach profile evolution.

## 2   Mathematical model

In the three phase flow solver sedInterFoam, mass and momentum conservation equations are solved for the dispersed phase
representing solid particles and the fluid phase constituted of the separated gas (air) and liquid (water) phases. The mass
conservation equations for the solid, liquid and gas phase are given by

$$\frac{\partial \phi}{\partial t} + \frac{\partial}{\partial x_i}[\phi u_i^s] = 0,\tag{1}$$

$$\frac{\partial}{\partial t}[\gamma(1-\phi)] + \frac{\partial}{\partial x_i}[\gamma(1-\phi)u_i^w] = 0,\tag{2}$$

$$\frac{\partial}{\partial t}[(1-\gamma)(1-\phi)] + \frac{\partial}{\partial x_i}[(1-\gamma)(1-\phi)u_i^g] = 0,\tag{3}$$

with $\phi$ the sediment volume concentration, $u_i^s$, $u_i^w$ and $u_i^g$ the solid, liquid and gas phase velocities, respectively, $x_i$ the position
vector with $i = 1, 2, 3$ representing the three spatial components and $\gamma$ the fluid indicator function defined as the ratio between
volume occupied by water and the total volume occupied by air and water in a cell ($\gamma = 1$ in water, $\gamma = 0$ in air and $0 < \gamma < 1$
at the interface); $\gamma$ is also commonly referred to the volume of fluid (Hirt and Nichols, 1981).

Instead of solving the three mass conservation equations, mass conservation can be fully accounted for by solving conser-
vation equations for $\phi$ and $\gamma$. Combining equations (1), (2) and (3) and assuming that $u_i^w = u_i^g$ at the interface because of the
no-slip boundary condition for velocities between air and water, we obtain the conservation equation for the indicator function
$\gamma$ following

$$\frac{\partial \gamma}{\partial t} + \frac{\partial}{\partial x_i}[\gamma u_i^f] - \gamma \frac{\partial u_i^f}{\partial x_i} = 0\tag{4}$$

with $u_i^f = \gamma u_i^w + (1-\gamma)u_i^g$ representing the velocity of the fluid phase constituted of the air and water. More details about the
derivation of equation 4 can be found in appendix A.

Similarly, instead of solving momentum conservation equations for the three phases, the air and water mixture is considered
as a single fluid with varying density and viscosity across the interface. The momentum conservation equations for the solid





and fluid phases are given by:

$$\frac{\partial}{\partial t}\left[\rho^s \phi u_i^s\right] + \frac{\partial}{\partial x_i}\left[\rho^s \phi u_i^s u_j^s\right] = \frac{\partial}{\partial x_i}\Sigma_{ij}^s + \phi \rho^s g_i + \phi f_i + M_i, \tag{5}$$


$$\frac{\partial}{\partial t}\left[\rho^f (1-\phi) u_i^f\right] + \frac{\partial}{\partial x_i}\left[\rho^f (1-\phi) u_i^f u_j^f\right] = \frac{\partial}{\partial x_i}\Sigma_{ij}^f + (1-\phi)\rho^f g_i + (1-\phi) f_i - M_i + \sigma \kappa \frac{\partial \gamma}{\partial x_i}, \tag{6}$$

with $\rho^s$ and $\rho^f = \gamma\rho^w + (1-\gamma)\rho^g$ representing the solid and fluid densities, $\rho^w$ and $\rho^g$ the water and air phases densities, $\Sigma_{ij}^s$ and $\Sigma_{ij}^f$ the solid and fluid phases effective stress tensors defined later in section 2.1, $g_i$ the gravitational acceleration, $f_i$ the momentum forcing term used to drive the flow, $M_i$ the momentum exchange term between the fluid and solid phases to be

described in section 2.2, $\sigma$ the surface tension coefficient and $\kappa$ the local air/water interface curvature.

Compared with the two-phase model sedFoam, only one additional unknown field $\gamma$ is added to the model, and therefore, only one additional equation for $\gamma$ (Eq. 4) needs to be solved. Also, the last term on the right hand side of the fluid phase momentum conservation equation (Eq. 6) needs to be included to model surface tension between air and water.

### 2.1 Effective stress tensors

The effective stress tensors of the fluid and solid phases are decomposed into normal and shear stresses following $\Sigma_{ij}^f = -P^f \delta_{ij} + T_{ij}^f$ and $\Sigma_{ij}^s = -P^s \delta_{ij} + T_{ij}^s$. Here, $P^f$ and $P^s$ denote the fluid and solid pressures respectively, $\delta_{ij}$ is the Kronecker symbol and $T_{ij}^f$ and $T_{ij}^s$ represent the fluid and solid shear stress tensors, respectively. These stress tensors are expressed in terms of the flow variables, which are defined as:

$$T_{ij}^f = \rho^f (1-\phi)\left[(\nu^{mix} + \nu_t^f)\left(\frac{\partial u_i^f}{\partial x_j} + \frac{\partial u_j^f}{\partial x_i} - \frac{2}{3}\frac{\partial u_k^f}{\partial x_k}\delta_{ij}\right) - \frac{2}{3}\lambda^f \delta_{ij}\right] \tag{7}$$

for the fluid phase and

$$T_{ij}^s = \rho^s \phi\left[(\nu^s + \nu_t^s)\left(\frac{\partial u_i^s}{\partial x_j} + \frac{\partial u_j^s}{\partial x_i} - \frac{2}{3}\frac{\partial u_k^s}{\partial x_k}\delta_{ij}\right) - \frac{2}{3}\lambda^s \delta_{ij}\right] \tag{8}$$

for the solid phase.

In these equations, $\nu^s$ denotes the solid phase viscosity, $\nu_t^f$ and $\nu_t^s$ are the fluid and solid phase eddy (turbulent) viscosities, $\lambda^f$ and $\lambda^s$ are the fluid and solid phases bulk viscosities and $\nu^{mix}$ is the mixture viscosity equal by default to the fluid phase

viscosity $\nu^f = \gamma\nu^w + (1-\gamma)\nu^g$ with $\nu^w$ and $\nu^g$ the liquid and gas phases viscosities or function of the sediment concentration when using the $\mu(I)$ rheology.

While air and water are considered Newtonian fluids having constant viscosities, the solid phase viscosity $\nu^s$ together with the solid phase bulk viscosity $\lambda^s$ and pressure $P^s$ are modeled using either the kinetic theory for granular flows or the $\mu(I)$ rheology. SedInterFoam is a direct extension of sedFoam documented in Chauchat et al. (2017). As a consequence, the solid

phase stress modelling in the two solvers is exactly the same. The reader is referred to Chauchat et al. (2017) for more details.

However, compared with the sedFoam version presented in Chauchat et al. (2017), the sedInterFoam user can select between using several choices of RANS turbulence models such as $k - \varepsilon$ or $k - \omega$ models (Chauchat et al., 2017; Nagel et al., 2020) or using a dynamic Lagrangian LES model introduced by Mathieu et al. (2021) to model the subgrid eddy viscosities.





## 2.2 Momentum exchange between the phases

The momentum exchange term $M_i$ between the two phases is composed of buoyancy, drag, lift, added mass forces and unresolved fluid-particle interaction forces $B_i$, $D_i$, $L_i$, $A_i$ and $I_i$, respectively. They are written in following expressions:

$$M_i = B_i + D_i + L_i + A_i + I_i \quad \text{with} \quad \begin{cases} B_i = -\phi \dfrac{\partial P^f}{\partial x_i} \\ D_i = \phi(1-\phi)K\left(u_i^f - u_i^s\right) \\ L_i = \phi(1-\phi)C_l \rho^m \|u_i^f - u_i^s\| \epsilon_{ijk} \dfrac{\partial u_k^m}{\partial x_j} \\ A_i = \phi(1-\phi)C_a \rho^f \left[ \dfrac{\partial u_i^f}{\partial t} + \dfrac{\partial u_i^f u_j^f}{\partial x_j} - \left( \dfrac{\partial u_i^s}{\partial t} + \dfrac{\partial u_i^s u_j^s}{\partial x_j} \right) \right] \\ I_i = -S_{US}(1-\phi)K\nu_t^f \dfrac{\partial \phi}{\partial x_i} \end{cases} \tag{9}$$

where $C_l = 0.5$ and $C_a = 0.5$ are the lift and added mass coefficients, $u_i^m = \phi u_i^s + (1-\phi)u_i^f$ and $\rho^m = \phi \rho^s + (1-\phi)\rho^f$ the mixture velocity and density respectively, $\varepsilon_{ijk}$ the Levi-Civita symbol, $S_{US} = 1/Sc$ is the inverse of the Schmidt number and
$K = \rho^s/t_s(1-\phi)$ the drag parameter with $t_s$ the particle response time modeled using a drag law. Several drag laws available in sedFoam are also present in sedInterFoam. As an example, the drag law proposed by Ding and Gidaspow (1990) combines the model of Ergun (1952) for high concentrations ($\phi > 0.2$) and the model of Wen and Yu (1966) for low concentrations ($\phi < 0.2$) following

$$t_s = \begin{cases} \rho^s \left( \dfrac{150\phi\nu^f \rho^f}{(1-\phi)d_p^2} + \dfrac{1.75\rho^f \|u_i^f - u_i^s\|}{d_p} \right)^{-1}, & \phi > 0.2 \\ \dfrac{4}{3} \dfrac{\rho^s}{\rho^f} \dfrac{d_p}{C_D \|u_i^f - u_i^s\|}(1-\phi)^{1.65}, & \phi < 0.2 \end{cases} \tag{10}$$

with $d_p$ the particle diameter, $C_D$ the drag coefficient given by

$$C_D = \frac{24}{Re_p}\left(1 + 0.15Re_p^{0.687}\right) \tag{11}$$

and the particle Reynolds number $Re_p$ is expressed as

$$Re_p = \frac{(1-\phi)d_p\|u_i^f - u_i^s\|}{\nu^f}. \tag{12}$$

The model proposed for the unresolved fluid-particle interaction term $I_i$ should only be included for RANS simulations.
Considering that no sub-grid interaction model has been rigorously validated in sedInterFoam for LES, $S_{US}$ is recommended to be set to zero. This term also plays a role in LES but is only important for particles having a diameter much smaller than the grid size. According to Ozel et al. (2013), it can be neglected when the grid size is on the order of the particle size.



## 3 Model implementation and solution algorithm

SedInterFoam is extended from the Eulerian two-phase model for sediment transport applications sedFoam
(https://github.com/sedFoam/sedFoam) (Chauchat et al., 2017; Mathieu et al., 2021) implemented using the open source CFD
library OpenFOAM (Jasak and Uroić, 2020). Mass and momentum equations are solved using a finite volume method and a
pressure-implicit with splitting of operators (PISO) algorithm is utilized for velocity-pressure coupling (Rusche, 2003).

As a comparison with sedFoam, the novelty resides in resolving the spatial and temporal evolution of free surfaces (air/water
interface) by solving equation (4) using the VOF method and the MULES algorithm (Rusche, 2003; Klostermann et al., 2013).
The idea behind the VOF method is to maintain $\gamma$ as a step function across the air/water interface and to ensure boundedness
($0 < \gamma < 1$). In other words, the interface has to be artificially compressed to balance numerical diffusion. To do so, we define
the artificial relative velocity between air and water $u_i^r = u_i^g - u_i^w$ and re-write equation (4) as:

$$\frac{\partial \gamma}{\partial t} + \frac{\partial}{\partial x_i}[\gamma u_i^f] - \gamma \frac{\partial u_i^f}{\partial x_i} + \frac{\partial}{\partial x_i}[\gamma(1-\gamma)u_i^r] = 0. \tag{13}$$

The last term of equation (13) only acts in the region $0 < \gamma < 1$ and the relative velocity $u_i^r$ is explicitly estimated to compress
the interface. More details about the numerical treatment of equation 13 are available in Klostermann et al. (2013).

After solving for the free surface evolution, fluid phase viscosity and density are updated and momentum conservation
equations for fluid and solid phases are solved as a two-phase system using the discretization procedure presented in Chauchat
et al. (2017). For free-surface wave applications, a popular toolbox called waves2Foam (Jacobsen et al., 2012) is included for
wave generation and absorption. The solution procedure is outlined as follows:

1. solve for indicator function $\gamma$ with interface compression method (Eq. 13)
2. update curvature $\kappa$, viscosity $\nu^f$ and density $\rho^f$
3. solve for sediment concentration $\phi$ (Eq. 1)
4. call waves2Foam library to update wave generation
5. update momentum exchange term (Eq. 9)
6. solve for solid phase stress using kinetic theory for granular flows or $\mu(I)$ rheology
7. solve for velocity-pressure coupling with PISO-loop
8. solve for turbulence closure (eddy viscosity for RANS or subgrid closure for LES)
9. go to the next time step

## 4 Model benchmarks and applications

In order to validate sedInterFoam, its implementation and to demonstrate its capability for challenging applications, four bench-
marks available as tutorials along with the source code are presented in this section. A dam-break tutorial case included in the
official OpenFOAM distribution is selected as the first benchmark to demonstrate the VOF capability of sedInterFoam (without
sediment) comparing with interFoam. The second benchmark is the sedimentation tutorials distributed with sedFoam to ver-



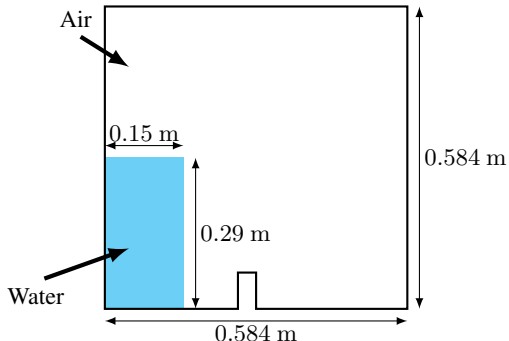

**Figure 1.** Sketch of the dam-break benchmark

ify the coupling between the solid and fluid phases. The third benchmark is a sheet flow configuration under monochromatic

non-breaking waves measured in large wave flume by Dohmen-Janssen and Hanes (2002) in order to validate the coupling

with waves2Foam for waves generation as well as the generation of sheet flows. Lastly, sedInterFoam is applied to simulate

a solitary wave plunging on an erodible sloping beach similar to the laboratory experiment of Sumer et al. (2011) in order to

validate the model for predicting beach profile evolution in the swash zone.

### 4.1   Dam-break

A 2D dam-break tutorial distributed with OpenFOAM is reproduced with sedInterFoam to validate the free-surface algorithm

without sediment.

A $0.15$ m $\times$ $0.29$ m water column initially at rest on the left side of a $0.584$ m $\times$ $0.584$ m tank otherwise filled with air is

released and impacts an obstacle at the center of the tank (figure 1). No turbulence model is used following the original tutorial

($i.e.$ $\nu_t^f = 0$). Liquid phase viscosity and density are specified to be $\nu^w = 1 \times 10^{-6}$ m$^2$.s$^{-1}$ and $\rho^w = 1000$ kg.m$^{-3}$ while the

gas phase viscosity and density are $\nu^g = 1.48 \times 10^{-5}$ m$^2$.s$^{-1}$ and $\rho^g = 1$ kg.m$^{-3}$. The surface tension coefficient is $\sigma = 0.07$.

The same numerical parameters are used for both interFoam tutorial case and present test case for a fair comparison. No-slip

boundary conditions are applied on the left, right and bottom walls and a the top boundary is free to the atmosphere and permits

both inflow and outflow. The mesh is composed of 9072 cells with grid size on the order of $4$ cm. An implicit first order scheme

(Euler) is used for time integration and second order upwind schemes are used for descretizing the convection terms. The time

step is adaptive and calculated to ensure a Courant-Friedrichs-Lewy (CFL) number lower than 1.

The comparison between sedInterFoam and interFoam at $0$ s, $0.3$ s and $0.6$ s are presented in figure 2. The time evolution of

the air/water interface (defined by $\gamma = 0.5$) calculated by sedInterFoam is in very good agreements with interFoam. The shape

and dynamics of the liquid phase impacting the obstacle is recovered and generation of air pockets downstream of the obstacle

are reproduced. The implementation of the free-surface evolution algorithm in sedInterFoam is therefore equivalent to that

in interFoam. Discrepancies appear only at $0.6$ s for which the flow becomes highly chaotic and originally small differences





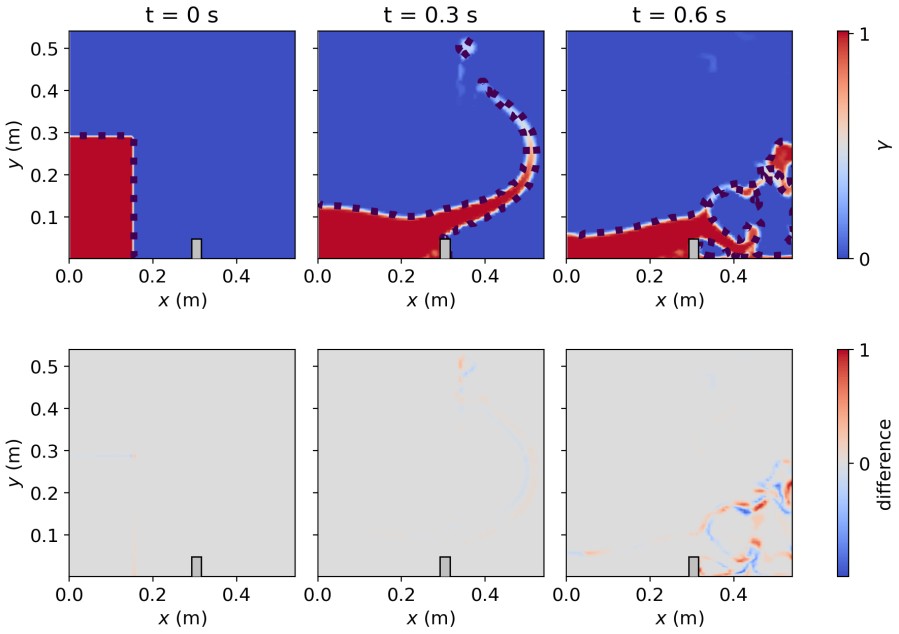

**Figure 2.** Comparison between $\gamma$ from sedInterFoam represented with the color map and interFoam with air/water interface represented by a dashed line (top panels) and difference between $\gamma$ field from sedInterFoam and interFoam (bottom panels) at 0 s (left panels), 0.3 s (middle panels) and 0.6 s (right panels) for the dam-break configuration

can lead to totally different results. Considering the numerical treatment of three phase flow equations, this behavior can be expected.

### 4.2 Sedimentation

The sedimentation benchmark taken from Chauchat et al. (2017) is reproduced with sedInterFoam but with an air-liquid interface specified at the top portion of the domain to compare with experimental data obtained by Pham-Van-Bang et al. (2008) and to validate the solid and liquid coupling through pressure-velocity algorithm, momentum exchange term and mass conservation.

The configuration consists of a suspension of mono-dispersed polystyrene particles of diameter $d_p = 290$ μm and density $\rho^s = 1050$ kg.m$^{-3}$ in Rhodorsil silicone oil having a density of $\rho^w = 950$ kg.m$^{-3}$ and viscosity $\nu^w = 1.02 \times 10^{-5}$ m$^2$.s$^{-1}$. The mixture is initially well-mixed at the bottom portion of the tank with a particle concentration $\phi = 0.5$ before sedimentation. The time evolution of the concentration is monitored in the experiment using a proton MRI device.

The numerical domain is 1DV decomposed into 240 cells over a depth of 12 cm. Compared with the sedFoam tutorial, an additional air layer with the same physical parameters as in the dam-break benchmark poresented in section 4.1 is added at the top of the domain to be able to validate mass conservation of the three phases throughout the simulation (figure 3). The time step is fixed to $\Delta t = 0.01$ s. A first-order implicit time-integration scheme (Euler) and a second order total variation diminish



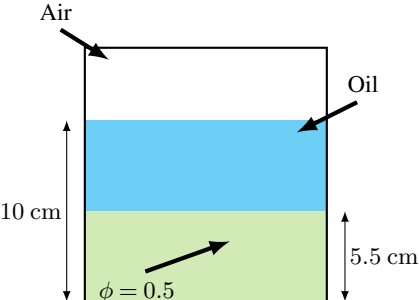

**Figure 3.** Sketch of the 1D sedimentation benchmark

(TVD) scheme for convection terms are used. A wall boundary condition is applied at the bottom and the pressure is fixed at the top to serve as a reference. The gradient of all other quantities is set to zero at the top boundary.

While settling, the sediment concentration profile shows two distinct interfaces. The upper interface corresponds to the transition between $\phi = 0$ and $\phi = 0.5$ while the lower interface corresponds to the transition between $\phi = 0.5$ to the maximum

concentration in the settled bed. A comparison of the temporal location of these two interfaces in the experiment and in the simulation is presented in panel (a) of figure 4. While, the concentration profiles at 232 s, 652 s, 1072 s and 1492 s are shown in panel (b) of figure 4. Very good agreements are observed with the measured data. For this benchmark, the performance of sedInterFoam is very similar to sedFoam reported in Chauchat et al. (2017).

The temporal evolution of the total volume fraction difference compared with initial value for the three phases are presented

in figure 5. Whereas the total volume fraction of particles is conserved throughout the simulation, some variation are observed for air and oil phases at the beginning. This is attributed to the time needed for the model to adapt from the initial conditions. After a few seconds, their total volume fractions no longer evolve. Overall, the total volume fraction differences for air and water phases remain lower than 0.5 percent. To conclude, the model and its implementation allow to reproduce pure sedimentation agreeing with measured data and the mass conservation is preserved well.

**4.3   Sheet flow under monochromatic waves**

To validate the coupling with the waves2Foam library for surface wave generation, an experimental configuration of sheet flow driven by monochromatic non-breaking waves from Dohmen-Janssen and Hanes (2002) is reproduced numerically with sedInterFoam. Cnoidal waves having a wave period of $T = 6.5$ s and wave height $H = 1.55$ m are generated and propagate over a bed composed of sand particles having a median diameter $d_p = 240$ μm and density $\rho^s = 2650$ kg.m$^{-3}$ in a 3.5 m

deep flume at the measurement section. Conductivity concentration meters buried in the sand measured the time series of the sediment concentration and allow for comparison with model results.

The numerical configuration used for this benchmark is the same as the configuration investigated by Kim et al. (2018). In Reynolds-average field, the flow is homogeneous in the spanwise direction and a 2D configuration is established. The



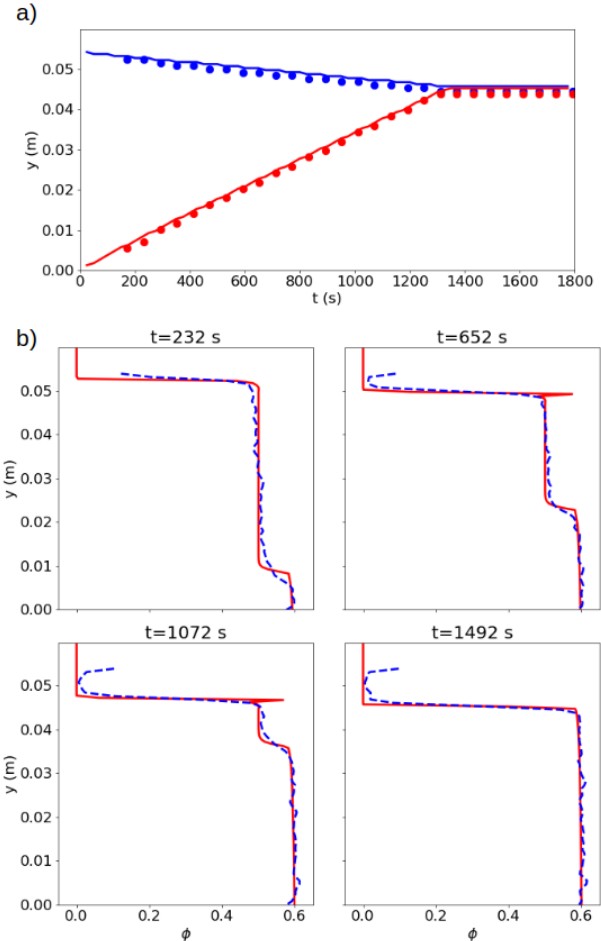

**Figure 4.** Temporal evolution of the upper (blue) and lower (red) interfaces in the pure sedimentation configuration (panel (a) with symbols for experimental results and solid line for simulation results) and concentration profiles at $t = 232$ s, 652 s, 1072 s and 1492 s (panel (b) with experimental results represented by the dashed blue line and simulation results represented by the solid red line).

numerical flume is 151.56 m in length and a 4 m long and 0.1 m deep sediment pit is located in the middle of the flume (figure

6). The mesh is generated with the OpenFOAM utilities blockMesh and snappyHexMesh. It is composed of around 2.85 million grid points with refined cells at the air/water and water/sediment interfaces. Details of the mesh close to the sand pit can be observed in figure 7. Wall boundary conditions are applied at the bottom while the top boundary is free to the atmosphere, wave are generated using a tenth-order stream function at the left hand side of the flume in the relaxation zone and a sponge-layer at the right hand side of the flume allows to absorb incoming waves and prevent reflection. The time step is adaptive to ensure a

maximum CFL number lower than 0.4. A second-order backward time integrator is used and a second-order TVD scheme is used for convection terms. Turbulence is modeled using the two-phase $k$-$\varepsilon$ model and the kinetic theory for granular flows is used to model the shear-dependent solid phase stresses.

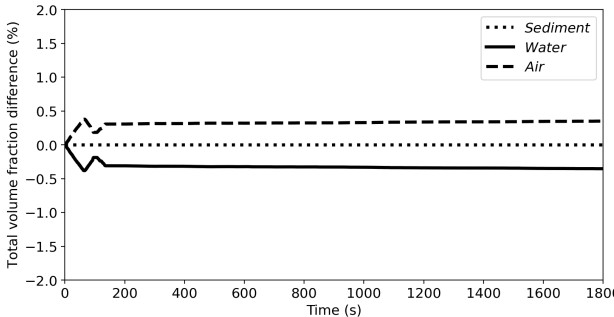

**Figure 5.** Time evolution of the total volume fraction difference for the solid, liquid and gas phase in the 1D sedimentation configuration.

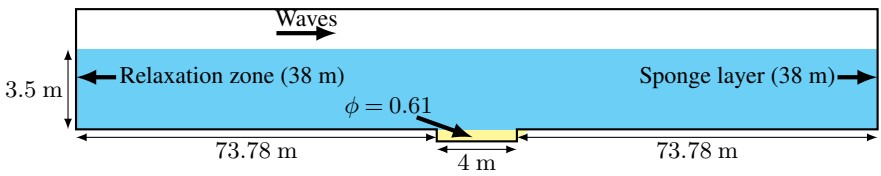

**Figure 6.** Sketch of the sheet flow configuration under monochromatic waves similar to the large wave flume experiment of Dohmen-Janssen and Hanes (2002).

To validate wave generation implementation through coupling with waves2Foam and the model capacity to predict sheet flow processes under surface waves, time series of the free stream velocity above the sediment bed and concentration profiles at wave crest, flow reversal and wave trough are compared to measured data in figure 8. The good agreement in the time series of free stream velocity highlights the capability of sedInterFoam to simulate non-linear surface wave processes and validates a proper implementation of waves2Foam.

For concentration profiles, numerical results are in good agreement with the measured data. Discrepancies can be observed at flow reversal for which suspended sediment are slightly over-predicted. Turbulence closure parameters controlling the equilibrium between settling and upward turbulent diffusion of sediment may be responsible for the over-predicted sediment suspension. A careful tuning of the model parameters would allow a more quantitative agreement between numerical and experimental results but is not in the scope of this study. The previous model sedWaveFoam yields almost identical results compared with sedInterFoam.

### 4.4 Plunging solitary wave

In order to demonstrate the novel capabilities of sedInterFoam for a more complex coastal application, an experimental configuration reported by Sumer et al. (2011) for a plunging solitary wave over an erodible bed is reproduced numerically. The solitary wave propagates in a 0.4 m deep flume and breaks on top of a beach of 1:14 slope composed of sand having a median





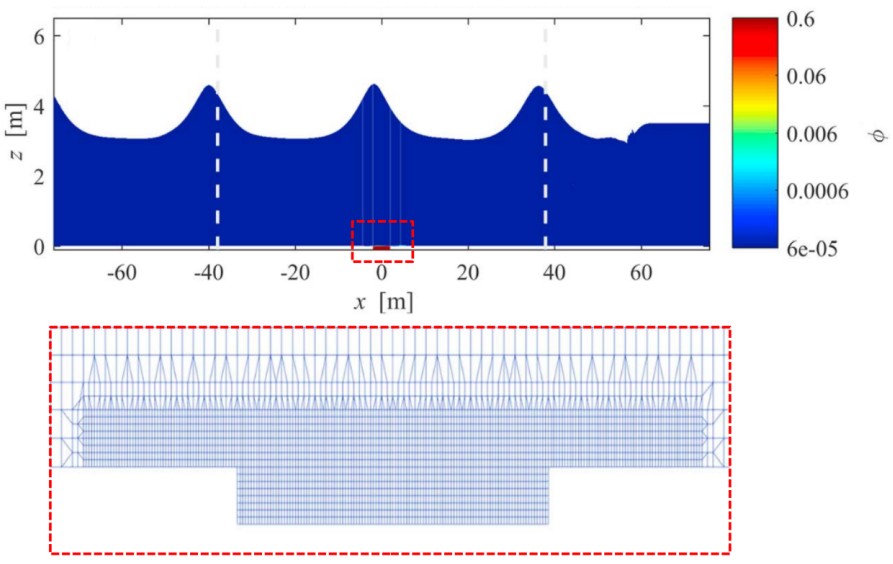

**Figure 7.** Snapshot of the sheet flow under monochromatic waves numerical configuration with air in white, water in blue and sediment in red (top panel) and details of the mesh (bottom panel). For visibility, the mesh is downsampled and the vertical scale is stretched with a factor 7. The dashed lines in the top panel correspond to the relaxation zones for wave generation on the left and sponge layer on the right.

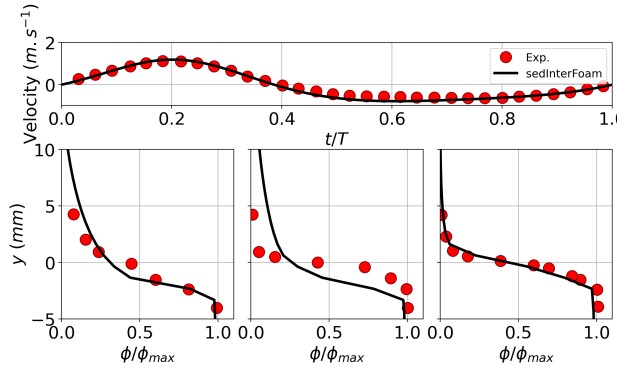

**Figure 8.** Time series of the free stream velocity above the sediment pit (top panel) and sediment concentration profiles at wave crest (bottom left panel), flow reversal (bottom middle panel) and wave trough (bottom right panel).

diameter of $d_p = 180$ μm. Sumer et al. (2011) recorded the beach profile evolution after the impact of four identical solitary waves. This is a critical test of sedInterFoam to simulate the challenging process of beach erosion in the swash zone.

Assuming spanwise homogeneity of the Reynolds-averaged flow, the numerical configuration is 2D. The left part of the flume is 12 m long with with a constant water depth of 0.4 m. The right portion of the flume consists of a 1:14 sloping beach loaded with sediment. The mesh is composed of 1.2 million grid points. The time-step is adaptive to ensure a maximum CFL



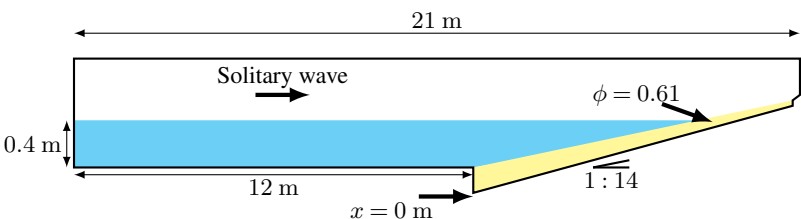

**Figure 9.** Sketch of the plunging solitary wave configuration similar to the laboratory experiment reported by (Sumer et al., 2011).

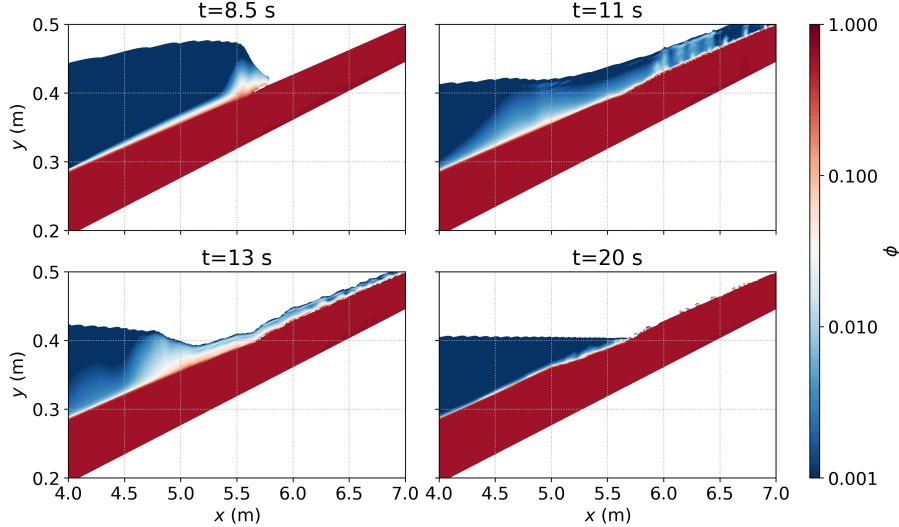

**Figure 10.** Snapshots of the sediment concentration $\phi$ at $t = 8.5$ s, 11 s, 13 s and 20 s from the simulation of the first solitary wave impacting the sloping beach. Local coordinate $x = 0$ m is defined at the toe of the sloping beach.

number lower than 0.3. The same numerical schemes as for the sheet flow under monochromatic waves configuration presented in section 4.3 were used. $k - \varepsilon$ turbulence model is used and solid phase stress is modeled using the $\mu(I)$ rheology.

Snapshots of the simulation showing the impact of the first solitary wave is presented in figure 10. At $t = 8.5$ s (top left panel of figure 10), the incoming solitary wave starts to break and erodes sediment at $x = 5.5$ m (the origin of the x-axis corresponds to the toe of the sloping beach). The morphological changes resulting from the passage of breaking wave at around $x = 5.6$ to 5.7 m are already visible at $t = 11$ s (top right panel of 10) which corresponds to the end of the uprush phase. During backwash at $t = 13$ s (bottom left panel of figure 10), a hydraulic jump reported by Sumer et al. (2011) is also observed between $4.5$ m

$5.2$ m under which sediment accretes. Eventually, at $t = 20$ s (bottom right panel of figure 10), the free surface returns at rest and the accretive pattern between $4.5$ m and $5$ m and the erosive pattern further landward become more evident.

Surface elevation numerically monitored at the toe of the beach ($x = 0$ m) and in the swash zone ($x = 4.63$ m, $4.87$ m, $5.35$ m and $5.85$ m) is compared to rigid-bed experimental data in figure 11. The shape of the incoming solitary wave is well

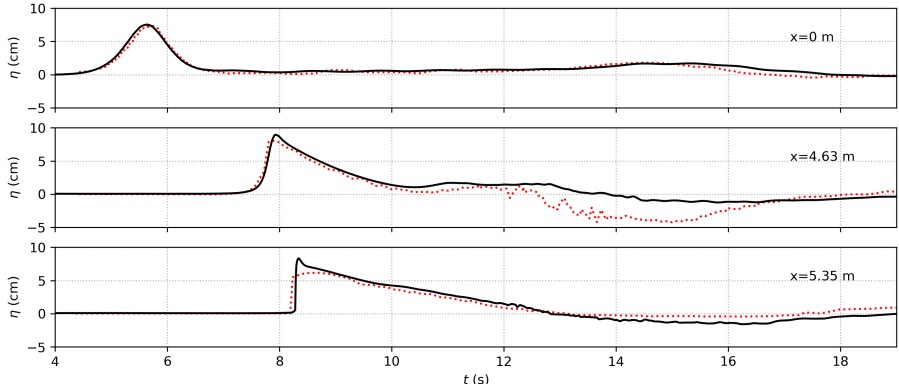

**Figure 11.** Free surface elevation at $x = 0$ m (toe of the beach), $x = 4.63$ m and $x = 5.35$ m from simulations with sedInterFoam (black solid lines) compared with rigid bed experimental data from Sumer et al. (2011) (red dotted lines).

captured by sedInterFoam at the different locations across the domain. At $x = 4.63$ m (second panel of figure 11), a significant
decrease of the water surface between $8$ s and $12$ s is under-estimated by the model. This time interval corresponds to the run
down phase when the newly formed hydraulic jump is migrating rapidly seaward while strong shear suspends sediment close
to the free surface. As a result, the differences observed in the free surface elevation may be the consequence of interactions
between sediment and water not present in the rigid bed experiments.

To compare the predictive capabilities of sedInterFoam, the evolution of the bed profile after four solitary waves is compared
to experiments in figure 12. Following the laboratory experiment, each successive solitary wave is sent after the previous
solitary wave impact to the flow field and bathymetry diminishes. The distinctive beach profile evolution features that can be
observed from the measured data are accretion between 4 m and 5 m corresponding to the location where a hydraulic jump
is observed during backwash and erosion between 5 m and 6 m corresponding to the intermittently wet and dry area of the
beach (*i.e.* upper swash). The position and amplitude of morphological changes in the erosion and accretion zone between 3 m
and 7 m are predicted fairly well by sedInterFoam. More experimental points would be required to conclude on the accuracy
of the model in the swash zone between 6 m and 7 m. However, the model does not reproduce the accretion zone located
at around 1 m corresponding to the formation of a sand bar. The processes involved in the generation of this morphological
feature most certainly include interaction between turbulence and sand particles in suspension. Such processes would require
better parameterization of RANS turbulence models or using a turbulence resolving simulations.

Overall, the presented simulation is a proof of concept that the morphodynamics in swash zone can be numerically simulated
using sedInterFoam. This new open-source model will allow to further investigate fine-scale hydromorphodynamics processes
to answer the many open-questions in this field.

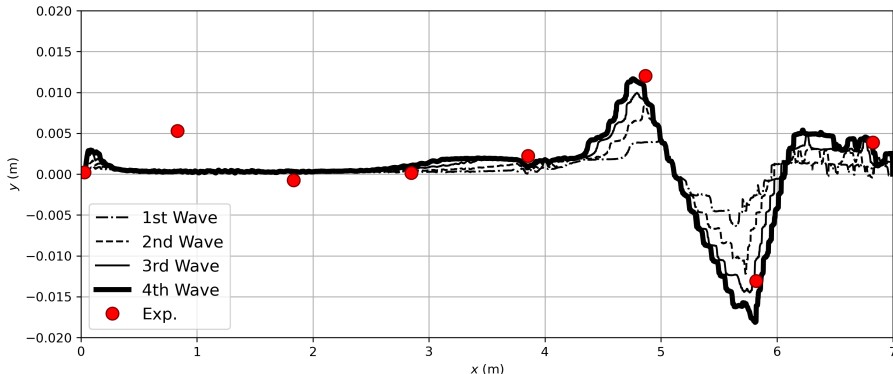

**Figure 12.** Morphological evolution of the sediment bed after four consecutive solitary waves from sedInterFoam simulation compared with experimental results from Sumer et al. (2011).

## 5   Conclusions

In this paper, sedIterFoam, an extension of the two-phase flow solver sedFoam (Chauchat et al., 2017) implemented using the open source library OpenFOAM (Jasak and Uroić, 2020) is presented. A third phase representing the air is included to model sediment transport applications driven by surface waves. The air/water interface is solved using the VOF method similar to that of interFoam (Rusche, 2003; Klostermann et al., 2013) but adapted for the present miscible liquid-solid phase. A coupling with the waves2Foam library (Jacobsen et al., 2012) is implemented to generate and absorb free surface waves. SedInterFoam includes all the previous features present in sedFoam such as turbulence models (RANS or LES) and solid phase stress models (kinetic theory for granular flows and $\mu(I)$ rheology).

The model has been successfully applied to a two-dimensional dam-break configuration to benchmark the numerical results with those produced by the existing OpenFOAM solver interFoam in order to verify the implementation of the free surface evolution algorithm. A comparison between sedInterFoam and sedFoam has been performed by simulating a one-dimensional sedimentation configuration in order to confirm that the implementation of a third phase did not affect mass conservation of each phase. A sheet flow configuration under monochromatic waves from (Kim et al., 2018) has been successfully reproduced indicating that the implementation of waves2Foam for wave generation/absorption is appropriate. Eventually, an experimental configuration of a solitary wave plunging on a sandy beach of 1:14 slope (Sumer et al., 2011) has been reproduced numerically to highlight the capabilities of sedInterFoam in simulating complex wave breaking, swash dynamics and beach profile evolution. Quantitative agreement between measured and simulated results are obtained particularly regarding the erosion and deposition process in the swash zone.

The development of the open source three-phase flow model sedInterFoam provides a modeling tool to investigate coastal sediment transport applications dominated by surface waves. While relatively computationally expensive, the physics-based model sedInterFoam can be a useful tool to gain insight into the complex physical processes associated with breaking waves,





sediment transport and morphodynamics and provide simulation data to improve empirical parameterization in regional-scale
morphodynamic models.

*Code availability.* sedInterFoam is distributed under a GNU General Public License v2.0 (GNU GPL v2.0) and is available on Zenodo at
https://zenodo.org/records/10577879 with the following DOI https://doi.org/10.5281/zenodo.10577879.

**Appendix A: Derivation of the indicator function transport equation**

In this section, the steps to derive the $\gamma$ transport equation are presented. Summing the mass conservation equations for the
solid, liquid and gas phases (Eq. 1, 2 and 3) and assuming $u_i^w = u_i^g = u_i^f$ because of the non-slip boundary condition at the
air/water interface allows us to write the mixture conservation equation as

$$\frac{\partial}{\partial x_i}[\phi u_i^s + (1-\phi)u_i^f)] = 0. \tag{A1}$$

Expanding equation (A1) and rearranging the terms gives

$$\frac{\partial}{\partial x_i}[\phi u_i^f] = \frac{\partial}{\partial x_i}[\phi u_i^s] + \frac{\partial u_i^f}{\partial x_i}. \tag{A2}$$

Starting from the liquid phase mass conservation equation (2), expanding it and using chain rule for the derivation of products
gives

$$\frac{\partial}{\partial t}[\gamma(1-\phi)] + \frac{\partial}{\partial x_i}[\gamma(1-\phi)u_i^f] = 0 \tag{A3}$$

$$\Rightarrow \frac{\partial \gamma}{\partial t} - \frac{\partial \phi \gamma}{\partial t} + \frac{\partial}{\partial x_i}[\gamma u_i^f] - \frac{\partial}{\partial}[\gamma \phi u_i^f] = 0 \tag{A4}$$


$$\Rightarrow \frac{\partial \gamma}{\partial t} - \phi\frac{\partial \gamma}{\partial t} - \gamma\frac{\partial \phi}{\partial t} + \frac{\partial}{\partial x_i}[\gamma u_i^f] - \phi\frac{\partial}{\partial x_i}[\gamma u_i^f] - \gamma u_i^f \frac{\partial \phi}{\partial x_i} = 0. \tag{A5}$$

Rearranging terms of equation (A5) allows to write

$$(1-\phi)\left[\frac{\partial \gamma}{\partial t} + \frac{\partial}{\partial x_i}[\gamma u_i^f]\right] - \gamma\left[\frac{\partial \phi}{\partial t} + u_i^f \frac{\partial \phi}{\partial x_i}\right] = 0 \tag{A6}$$

$$\Rightarrow (1-\phi)\left[\frac{\partial \gamma}{\partial t} + \frac{\partial}{\partial x_i}[\gamma u_i^f]\right] - \gamma\left[\frac{\partial \phi}{\partial t} + \frac{\partial}{\partial x_i}[\phi u_i^f] - \phi\frac{\partial u_i^f}{\partial x_i}\right] = 0 \tag{A7}$$



Replacing the second to last term of equation (A7) by equation (A2) and remembering the solid phase mass conservation equation (1) reads

$$(1-\phi)\left[\frac{\partial \gamma}{\partial t}+\frac{\partial}{\partial x_i}[\gamma u_i^f]\right]-\gamma\left[\underbrace{\frac{\partial \phi}{\partial t}+\frac{\partial}{\partial x_i}[\phi u_i^s]}_{=0}+\frac{\partial u_i^f}{\partial x_i}-\phi\frac{\partial u_i^f}{\partial x_i}\right]=0 \tag{A8}$$

Rearranging the terms allows to obtain the final form the $\gamma$ transport equation following

$$\Rightarrow (1-\phi)\left[\frac{\partial \gamma}{\partial t}+\frac{\partial}{\partial x_i}[\gamma u_i^f]\right]-\gamma\frac{\partial u_i^f}{\partial x_i}+\gamma\phi\frac{\partial u_i^f}{\partial x_i}=0 \tag{A9}$$

$$\Rightarrow (1-\phi)\left[\frac{\partial \gamma}{\partial t}+\frac{\partial}{\partial x_i}[\gamma u_i^f]-\gamma\frac{\partial u_i^f}{\partial x_i}\right]=0 \tag{A10}$$

$$\Rightarrow \frac{\partial \gamma}{\partial t}+\frac{\partial}{\partial x_i}[\gamma u_i^f]-\gamma\frac{\partial u_i^f}{\partial x_i}=0. \tag{A11}$$

*Author contributions.* Model development: AM, YK, CB, JC. Validation cases development: AM, YK, CB, JC, TH. Project management: YK, JC, TH. Manuscript preparation : AM with contribution from all co-authors.

*Competing interests.* The contact author has declared that none of the authors has any competing interests.

*Acknowledgements.* This study was supported by NSF (CMMI-2050854; OCE-2242113), ONR (N00014-22-1-2412) and SERDP (Strategic Environmental Research and Development Program MR-1478). Yeulwoo Kim is supported by the Basic Science Research Program through
the National Research Foundation of Korea (NRF) funded by the Ministry of Education (NRF-2021R1F1A1062223) and the Development of Advance Technology for Ocean Energy, Harbor and Offshore Structure project (PEA-0131) funded by the Korea Institute of Ocean Science and Technology (KIOST). Numerical simulations presented in this study benefited from computing resource provided by the DARWIN and Caviness clusters at the University of Delaware and the EXPANSE cluster at San Diego Supercomputer Center via ACCESS.





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
