# Peer review of "sedInterFoam 1.0: a three-phase numerical model for sediment transport applications with free surfaces"

_Geoscientific Model Development, 2024_

## Referee Comment (RC2)

**General comments:**

The presented numerical study titled '*sedInterFoam 1.0: a three-phase numerical model for sediment transport applications with free surfaces*' examines four simulations while comparing the performance of the updated sedInterFoam, modified after sedFoam, interFoam, and waves2Foam, in simulating three-phase flows, and eventually targeting sediment transport modeling under coastal environments. The work is appreciable and the manuscript is written quite well. I hope the following comments and suggestions are helpful in improving the manuscript:

**Specific comments:**

1. I miss a grid sensitivity test and the effect of turbulence model, which could improve the obtained results, e.g., the disagreements discussed in lines 246-248.
2. Fig. 2: compare the volume fraction contour plot and air/water interphase separately.
3. The interfaces shown in Fig. 4(a) correspond to what exact phi values?
4. Fig. 4: please discuss about the discrepancies observed at y around 0.05 m for t = 652 s and later.
5. Why separate limiting CFL were used in different cases. What was the basis behind the selected values? Please discuss.
6. Line 247: please also plot the sedWaveFoam results in Fig. 8.
7. Line 253: 0.4 m deep flume or the flow depth is 0.4 m?
8. Fig. 8: stretch the top panel of Fig. 8 vertically. Show error bars and discuss about % errors in the simulated results. Also, place the locations of the profiles in Fig. 8 (bottom panels) in Fig. 7 for a better understanding. Define *y*.
9. Fig. 10: Fig. 10: why phi is up to 1.0 since phi of sediment is 0.61. Also, what are the phi values of the interfaces?
10. Fig. 10: were there no experimental data to compare the results? Zoom into the areas of erosion, deposition etc. for a better visibility and understanding of the findings discussed in lines 265-264.
11. Fig. 11: wave profiles in Fig. 11 are significantly lower than the water surface profiles provided in Fig. 10. Please cross check. Use m in *y* axis too.

**Technical corrections:**

1. 'modeling' in place of 'modelling'. Please follow either US English or UK English, do not mix.

2. Which version of OpenFOAM the code modifications are based on? Please mention in the manuscript.

3. Please provide the full form of 1DV.

4. Use 'Fig. 4(a)' in place of 'panel (a) of figure 4'. Follow the same at other locations throughout the manuscript.

5. Line 240: use 'compared with' in place of 'compared to'.

6. Details of the solitary wave characteristics are missing.

7. Section 4.4, Fig. 9: mention the depth of the sand layer.

8. Fig. 7: what is $z$ in Fig. 7. Should it be $y$?

9. Please mention whether the simulations were run in parallel. How many cores were used and what were the simulation execution/run time.

10. Line 265: please recheck if 'sediment accretes' fits here.

11. Line 267, Fig. 11: 4.87 m and 5.85 m results are missing in Fig. 11.

12. Lines 275-276: 'Following the laboratory experiment, each successive solitary wave is sent after the previous solitary wave impact to the flow field and bathymetry diminishes' – provide time intervals between the successive waves.

13. Provide % errors correspond to the highest deposition and deeper scouring points in Fig. 12.

14. LES not tested in the study but mentioned many times. In the future, do you plan to extend this study using LES?

---

## Author Response (AR1)

**Referee #1**

Line 77. In my opinion, the equations are derived not only for air and water. Although this is not a problem, the authors should use u_l/u_g or u_a/u_w consistently in the equations. At least, the authors should use $u_i^a$ in Equation 3 because Equation 2 has already used $u_i^w$. It may be better to replace air/water using gas/liquid throughout the text, but this is just a suggestion.

**Response :** We would like to thank the reviewer for this great suggestion. We agree that the notation used in the paper should be more general and in the revision "air:water" is replaced by "gas/liquid" throughout the text.

Line 95. How is the curvature calculated using VOF? How is it implemented in the model? Why is the gravitational acceleration different for two fluid phases?

**Response** : The curvature is calculated as the divergence of the unit vector normal to the surface. More details can be found in Klosterman et al. (2013) and this citation is added in line 96. The gravitational acceleration $g_i = 9.81 m^2/s$ is the same for all three phases, but in the momentum equation, each term is expressed as force per unit volume and hence the gravitational term is weighted by the corresponding density and volumetric concentration.

By solving this equation, does it mean u_f is solved altogether for the fluid phase, and then u_gas and u_liquid are solved separately by using the mass conservation equation? If doing so, it seems that gamma and u_f are only determined by the initial and boundary conditions. Does it mean air and water move together and not decoupled? I think this is also the problem with the interFoam solver.

**Response :** As mentioned on line 80, there is a no-slip boundary condition between gas and liquid. Indeed, the gas and liquid phase are not decoupled. As a consequence, u_gas and u_liquid are assumed equal at the interface, defined over the whole numerical domain and related to u_f through gamma which is solved using the indicator function conservation equation (4). Gamma and u_f are determined by initial and boundary conditions, interaction forces with the sediment phase and body forces such as gravity. The purpose of this study is to expand the two-phase model SedFoam with an air-water interface tracking capabilities and hence we essentially couple interFoam with SedFoam and used all the assumptions in the interFoam.

Because the interFoam solver does not decoupled the air/water, in my past experiments I found that the volume fraction gamma will decay in water and increase in air. Have the authors found this in the simulations? Are there any issues with the 0.99, 0.95, or 0.90 contours of the volume fractions?

**Response :** Many thanks for pointing out this issue. So far the authors did not witness such behavior of the indicator function gamma. No issues with contours of the volume fraction can be reported. The interface compression step in the algorithm should prevent this behavior.

Are the stress tensors for the fluid phase treated the same as the fluid phase for interFoam solver?

**Response :** The stress tensors for fluid and solid phases are treated as in the sedFoam solver and please refer to Chauchat et al. (2017) for more information.

Line 120. I think the gravitational force is already considered in the momentum equations. The B term should be the pressure gradient force, not buoyancy.

**Response :** In the two-phase flow formalism, buoyancy force between fluid and solid phase is accounted for through the fluid pressure gradient as expressed in equation (9). The gravitational force is the solid phase momentum equation only takes into account the solid phase contribution. This can be better understood by summing momentum equations for both phases to get the mixture momentum equation. More detailed can be found in Jackson (2000).

In Equation 10, I am not sure if t_s is continuous at phi = 0.2 when C_D is determined by the Reynolds number.

**Response :** t_s is not continuous at phi=0.2 and can cause numerical issues in rare cases. Most of the time, considering that the interface between clear fluid (phi=0) and sediment bed (phi=phi_max) is very sharp, there is *de facto* a step function in the drag coefficient.

Line 136. Does it mean the fluid interaction force is zero when using LES?

**Response :** Thanks for raising the good point. It means that the unresolved (sub-grid) interaction force is zero when using LES. As mentioned in the paragraph (line 135), this is a reasonable assumption for grid size on the same order as the particle size (Ozel et al., 2013).

Line 155. I think part 4 are not always used in the simulations. Same for the turbulence models.

**Response :** Reviewer 1 is right, this is only used for surface wave applications; the steps are tagged as optional in the revised manuscript

For the dam break case, are you using the example case from InterFoam solver? Any references?

**Response :** Yes the dam-break case is the example case from interFoam solver but the authors could not find any references for this case. This has been mentionned in lines 167 and 176 of the revised manuscript.

Line 202. Does 1DV mean 1-dimension?

**Response :** 1DV means 1-dimension vertical. This has been clarified in the revised manuscript

Figure 4. Why does the upper interface (blue) have higher volume fraction at t = 652/1072/1492? Why does the lower interface (red) have saturated volume fraction on the top at t = 652/1072? Is this because of numerical issues when determining the interface?

**Response :** The dashed-blue curve represents experimental results and the higher volume fraction a the top comes from measurements. The saturated volume fraction for the red (numerical curve) comes from numerical oscillations at the interface because of the sharp concentration gradient. This behavior has also been observed with sedFoam solver and results from the difficulty to handle the propagation of a shock by the mass conservation equations.

Figure 4. For Panel (a) there are also ups and downs for the interface. Is this also because of some numerical issues?

**Response :** The ups and downs in panel (a) is only the results of spatial discretization. The position of the interface is determined at the center of the cell. When the interface crosses the boundary between two cells, there is a "jump" in the curve. Rather than ups and downs, the curve shows a step behavior.

Line 217. The volume fraction for air/water still evolves, but at a very small rate.

**Response :** Thanks for pointing this out. It has been mentioned that air/water evolves at a smaller rate in the revised manuscript.

Figure 7. Is this case also aimed to demonstrate the model for unstructured grid?

**Response :** Reviewer 1 is right, this case also aims to demonstrate the model capabilities on unstructured grids. It has been mentioned in the revised manuscript.

Figure 12. Why is the experiment result at x = 1m different from others? Any comments?

**Response :** At 1m, the sediment eroded in the swash zone is transported in the offshore direction and accumulates and forms a sand bar. This process is a result of a complex interplay between wave-breaking induced current (undertow) and turbulence-particle interactions. It is mentioned in the manuscript line 283 : "The processes involved in the generation of this morphological feature most certainly include interaction between turbulence and sand particles in suspension. Such processes would require better parameterization of RANS turbulence models or using a turbulence resolving simulations." Since the purpose of this work is to present the capability newly developed model for various benchmarks and applications, more detailed model investigation on turbulence closure and suspended load transport will be deferred to future work.

For test case 3 and 4, the authors demonstrated that the implemented model performed good predictions, but it is hard to tell the improvement after considering the interaction between air, water, and sediment. How is the effect of the air-sea interface in case 3? How does the morphological elevation in case 4 change the wave run-up?

**Response :** For case 3, there is no direct interaction between the three phases but having real waves allows to reproduce physical phenomenon known as wave boundary layer streaming. It allows to reproduce more realistic cross-shore sediment transport under surface waves without relying on simplifying assumptions such as sediment transport in oscillating water tunnels. For case 4, the definition of the swash zone implies the presence of air, water and sediment due to the intermittent wetting and drying of the beach and it must be simulated with the present model (cannot be simulated using previous SedFoam) . The reviewer raised a good point. The change in bed morphology in case 4 is small and does not affect the wave run-up after 4 waves (our ongoing work show simulating an immobile and impermeable bed using InterFoam show very similar run up). However, if more waves were impacting the beach, the morphology would eventually impact wave run-up. Investigating this important issue is our ongoing and future work.

**Referee #2**

**Specific comments:**

1.I miss a grid sensitivity test and the effect of turbulence model, which could improve the obtained results, e.g., the disagreements discussed in lines 246-248.
**Response** : Reviewer 2 is right, results may be sensitive to grid resolution and turbulence model choices. It is mentioned in the manuscript that results may be sensitive to turbulence closure parameters. The authors believe that sensitivity to grid resolution and turbulence closure are out of the scope of the present study in the description of the test case but have been mentioned in the revised manuscript.

2. Fig. 2: compare the volume fraction contour plot and air/water interphase separately.
**Response**: The authors have tried before to plot separately volume fraction (from interFoam) and air/water interphase (from sedInterFoam) but comparison between the two models was impossible because differences between sedInterFoam and interFoam are almost indiscernible. The authors therefore opted to plot them on the same figure and add a colormap representing the error between the two models.

3. The interfaces shown in Fig. 4(a) correspond to what exact phi values?
**Response**: In the manuscript, it is mentioned that (line 210) the upper interface corresponds to the transition between phi=0 and phi=0.5 while the lower interface corresponds to the transition between phi=0.5 to the maximum concentration in the settled bed. Exact values taken to monitor the interfaces are phi=0.25 for the upper interface and 0.55 for the lower interface.

4. Fig. 4: please discuss about the discrepancies observed at y around 0.05 m for t = 652 s and later.
**Response**: The discrepancy observed comes from numerical oscillations at the interface because of the sharp concentration gradient. This behavior has also been observed with sedFoam solver and results from the difficulty to handle the propagation of a shock by the mass conservation equations. This discussion is added in line 215.

5. Why separate limiting CFL were used in different cases. What was the basis behind the selected values? Please discuss.
**Response**: The different cases are taken from multiple sources (interFoam tutorial cases, sedFoam tutorial cases, etc) and default values used in the original cases were used. Choice of the CFL number is the result of a compromise between stability and computational time.

6. Line 247: please also plot the sedWaveFoam results in Fig. 8.
**Response**: Results from sedWaveFoam were added.

7. Line 253: 0.4 m deep flume or the flow depth is 0.4 m?
**Response**: Thanks for raising this mistake. Flow depth is 0.4m. Correction has been made in the revised manuscript.

8. Fig. 8: stretch the top panel of Fig. 8 vertically. Show error bars and discuss about % errors in the simulated results. Also, place the locations of the profiles in Fig. 8 (bottom panels) in Fig. 7 for a better understanding. Define y.
**Response**: Top panel of figure 8 has been stretched. Unfortunately, the authors do not have information about measurement errors but differences have been discussed in the revised manuscript. It is also mentioned that the profiles shown on figure 8 represent concentration in the middle of the sediment pit. Y is the vertical coordinate.

9. Fig. 10: Fig. 10: why phi is up to 1.0 since phi of sediment is 0.61. Also, what are the phi values of the interfaces?
**Response**: The maximum concentration is not 1 but the color bar goes up to 1 to avoid saturated colorbar in the plot. This has been clarified in the revised manuscript.

10. Fig. 10: were there no experimental data to compare the results? Zoom into the areas of erosion, deposition etc. for a better visibility and understanding of the findings discussed in lines 265-264.
**Response**: Reviewer 2 raises a good point mentioning the importance of experimental results to compare to and having better visibility on the areas of erosion/deposition. Figure 11 show

comparison with experimental results for the free surface evolution and figure 12 shows comparison with experimental results for the morphological evolution. Vertical axis is stretched for better visibility on zones of erosion and deposition. Results are discussed in the paragraph starting at line 274

11. Fig. 11: wave profiles in Fig. 11 are significantly lower than the water surface profiles provided in Fig. 10. Please cross check. Use m in y axis too.
**Response**: Variation of the free surface elevation are comparable between figure 10 and 11 with variation on the order of 0.05m to 0.1m in magnitude. Meters have been used in the y axis in the revised manuscript.

**Technical corrections:**

1. 'modeling' in place of 'modelling'. Please follow either US English or UK English, do not mix.
**Response**: 'modeling' has been used in the revised manuscript.

2. Which version of OpenFOAM the code modifications are based on? Please mention in the manuscript.
**Response**: the present model has been used with OpenFOAM-v2106 and OpenFOAM-v2112. This has been clarified in the revised manuscript

3. Please provide the full form of 1DV.
**Response**: 1DV means 1-dimension vertical. This has been clarified in the revised manuscript.

4. Use 'Fig. 4(a)' in place of 'panel (a) of figure 4'. Follow the same at other locations throughout the manuscript.
**Response:** Thank you for the suggestion. The recommendation has been followed across the revised manuscript.

5. Line 240: use 'compared with' in place of 'compared to'.
**Response**: 'compared with' has been used in place of 'compared to' in the revised manuscript.

6. Details of the solitary wave characteristics are missing.
**Response**: The authors believe that adding the solitary wave characteristics may overload the present manuscript. However, reference to the original experimental paper has been added in which all information required are presented.

7. Section 4.4, Fig. 9: mention the depth of the sand layer.
**Response**: Depth of the sand layer is 0.15m at the toe and 0.05m at the top of the beach. This has been mentioned in section 4.4 of the revised manuscript.

8. Fig. 7: what is z in Fig. 7. Should it be y?
**Response**: Thank you for pointing this inconsistency. Indeed, vertical coordinate is y in this context. This has been corrected in the revise manuscript.

9. Please mention whether the simulations were run in parallel. How many cores were used and what were the simulation execution/run time.
**Response**: information about number of cores and computational time has been added for the different cases in the corresponding sub-sections.

10. Line 265: please recheck if 'sediment accretes' fits here.
**Response**: 'sediment accretes' has been replaced by 'sediment accumulates'.

11. Line 267, Fig. 11: 4.87 m and 5.85 m results are missing in Fig. 11.
**Response**: Thanks for pointing that mistake, which is corrected in the revision. A previous version of the figure included results at 4.87m and 5.85m but overloaded the manuscript without adding more information. The authors decided to only include results at 4.63m and 5.35m.

12. Lines 275-276: 'Following the laboratory experiment, each successive solitary wave is sent after the previous solitary wave impact to the flow field and bathymetry diminishes' –
provide time intervals between the successive waves.
**Response**: Successive solitary waves are sent after the free-surface becomes still again. In the case of the simulations presented in the manuscript, it corresponds to intervals of 20 seconds. This has been clarified in the revised manuscript.

13. Provide % errors correspond to the highest deposition and deeper scouring points in Fig. 12.
**Response**: % of error have been mentioned in the revised manuscript.

14. LES not tested in the study but mentioned many times. In the future, do you plan to extend this study using LES
**Response**: Thanks for the suggestion. sedInterFoam contains the capabilities implemented in sedFoam. Since sedFoam includes LES, sedInterFoam can be used in LES studies as well. Many applications can benefit from LES in their investigations. SedInterFoam is currently used in sediment transport studies in the swash zone using LES.